# Global CO$_2$ emissions from dry inland waters share common drivers across ecosystems

P. S. Keller ⓘ et al.[#]

Many inland waters exhibit complete or partial desiccation, or have vanished due to global change, exposing sediments to the atmosphere. Yet, data on carbon dioxide (CO$_2$) emissions from these sediments are too scarce to upscale emissions for global estimates or to understand their fundamental drivers. Here, we present the results of a global survey covering 196 dry inland waters across diverse ecosystem types and climate zones. We show that their CO$_2$ emissions share fundamental drivers and constitute a substantial fraction of the carbon cycled by inland waters. CO$_2$ emissions were consistent across ecosystem types and climate zones, with local characteristics explaining much of the variability. Accounting for such emissions increases global estimates of carbon emissions from inland waters by 6% (~0.12 Pg C y$^{-1}$). Our results indicate that emissions from dry inland waters represent a significant and likely increasing component of the inland waters carbon cycle.

[#]A full list of authors and their affiliations appears at the end of the paper.

Both natural and human-made inland waters are frequently impacted by drying[1–3]. Such ecosystems may partially or fully desiccate temporarily, and in some cases inland waters have even desiccated permanently[4,5]. Drying can result from natural hydrological factors (e.g. snowmelt driven lake-level fluctuations[6], or the seasonal desiccation of intermittent streams or rivers[7]) or from anthropogenic factors[8] (e.g. agricultural diversions, or water level fluctuation in reservoirs[9]). Indeed, climate change and increased water abstraction are together expected to exacerbate the widespread prevalence of dry inland waters[10]. Two-thirds of the planet's first-order mid-latitude (below 60°) streams are estimated to flow only temporarily, as are one-third of larger, fifth-order rivers[11]. Furthermore, seasonal desiccation affects 18% (~800,000 km²) of the global surface area covered by inland waters, exposing previously submerged sediments to the atmosphere[10]. Such hydrologically dynamic environments are typically excluded from inland aquatic carbon (C) budgets and not explicitly accounted for in the terrestrial budgets, representing a potential blind spot in global C cycling estimates[12]. In accordance with previous work[12], we define dry inland waters as the areas of lotic and lentic aquatic ecosystems on the Earth's land masses where surface water is absent, and sediments are exposed to the atmosphere.

Gaseous C emissions from inland waters to the atmosphere play an important role in the global C cycle[11,13–15]. However, recent studies have shown that exposed sediments following the desiccation of inland waters can contribute $CO_2$ emissions to the atmosphere at greater rates than those measured from the water surface during inundated periods[16–18]. Initial estimates predicted that these emissions may be relevant at a global scale[12,19]. Specifically, if the fluxes from desiccated areas were added to existing global estimates of $CO_2$ emissions from inland waters[11,20,21] they would result in 0.4–10% higher estimates of inland $CO_2$ emissions to the atmosphere. However, these emission estimates from desiccated areas were based on a small number of localised studies, and convincing evidence for the global importance of this pathway is still lacking. Many inland water ecosystems are affected by water diversion, water abstraction and climate change[8,22], leading to likely future increases in exposed sediment areas. Therefore, there is an urgent need to quantify the global $CO_2$ emission from dry inland waters and to deepen our understanding of the environmental factors regulating them.

We hypothesised that $CO_2$ emissions from dry inland waters are above reported mean aquatic rates, thus making emissions from dry inland waters globally relevant. We further hypothesised that sediment-atmosphere emissions vary as a function of parameters controlling $CO_2$ production rates (such as organic matter supply, temperature and moisture) and parameters controlling the transport of gas to the atmosphere (e.g. sediment texture) as well as geographical properties of the sampling locations, which influence the biogeochemical conditions. To test these hypotheses, we conducted a global survey in which we quantified $CO_2$ fluxes from 196 dry inland waters distributed across all continents except Antarctica, representing diverse inland water ecosystem types (rivers, lakes, reservoirs and ponds) and climate zones (tropical, arid, temperate, continental and polar). We compared the magnitude of these fluxes to those measured at adjacent uphill soils as well as global estimates for inundated water bodies compiled from the literature. To investigate potential drivers, we modelled the influence of environmental variables on the magnitude of $CO_2$ emissions from the sediments to the atmosphere. Because dry inland waters are environments in between aquatic and terrestrial ecosystems, we aimed to disentangle whether $CO_2$ emissions from dry inland waters were closer in value to those from aquatic or terrestrial ecosystems to improve the accuracy of current upscaling models of global $CO_2$ emissions.

## Results

**Magnitude of $CO_2$ emissions from dry inland waters.** Sediment $CO_2$ fluxes ranged from −27 to 2968 mmol m⁻² d⁻¹ (mean ± SD = 186 ± 326, median = 93, $n = 196$, Fig. 1; negative values indicate a net flux from the atmosphere to the sediments). This study provides the first data confirming that elevated $CO_2$ emissions from desiccated sediments reported in prior localised studies[17,19] (Supplementary Table 1) are globally prevalent and an intrinsic characteristic of dry inland waters. The sampled sites include a great diversity of environmental conditions (Fig. 1), although the collaborative nature of the study precluded an even geographical distribution of sampling efforts, and sites in the temperate zone dominate the dataset. Measured $CO_2$ emissions from dry inland waters to the atmosphere were an order of magnitude higher than average water surface emissions (water-to-atmosphere)

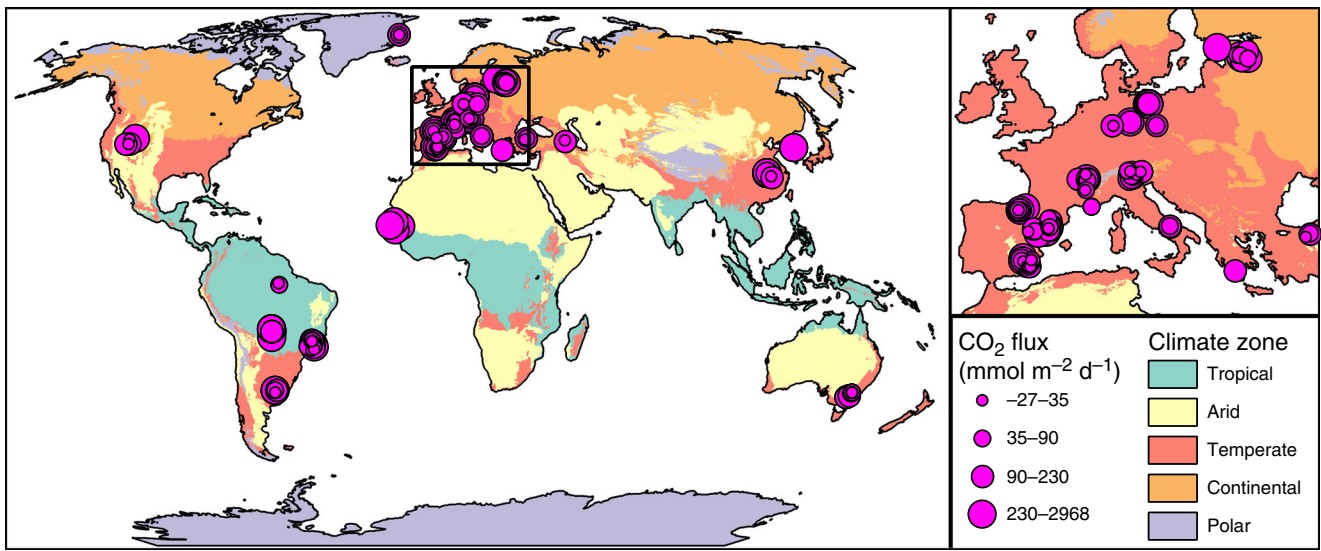

**Fig. 1 Global distribution of $CO_2$ fluxes from dry inland waters.** Size of pink dots indicates magnitude of measured $CO_2$ fluxes. Background colours indicate climate zones according to the Köppen–Geiger climate classification system[52]. Inset illustrates the spatial distribution within the most densely sampled area.

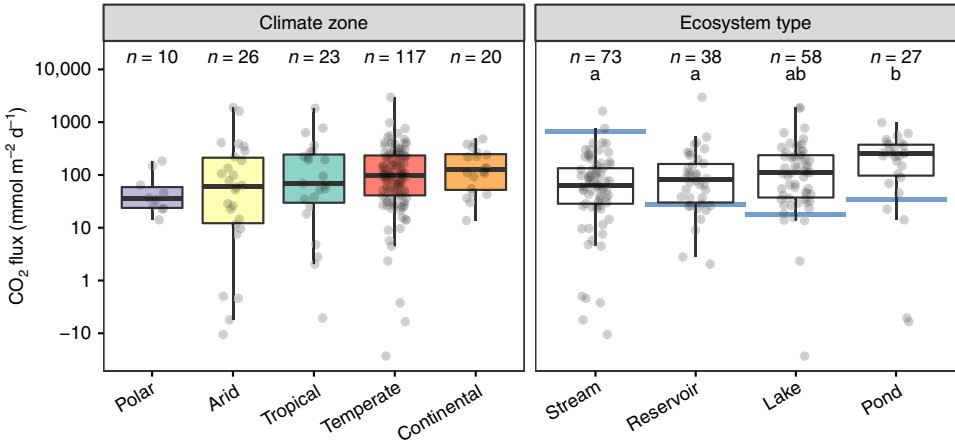

**Fig. 2 CO$_2$ fluxes separated by climate zones and ecosystem types.** Box = 25th and 75th percentiles, whiskers = 1.5* inter-quartile range. Black line = median. Blue lines represent average estimates of CO$_2$ emissions for inland waters as reported in the literature[11, 20, 21]. Colours refer to climate zones as defined in Fig. 1. Note that the y-axis is presented on a log$_{10}$ scale to show a wide range of flux values. Letters indicate significant differences between ecosystem types (Kruskal–Wallis test and Dunn's post hoc test, $P < 0.05$).

previously reported for lentic waters (27 mmol m$^{-2}$ d$^{-1}$), but lower than average emissions reported for lotic waters (663 mmol m$^{-2}$ d$^{-1}$) (Fig. 2; Supplementary Table 1).

Higher CO$_2$ emissions to the atmosphere from exposed sediments relative to lentic inland water surface emissions are likely due to a closer coupling of CO$_2$ production and gas flux in dry sediments (due to the lack of an intervening layer of water) as well as increased CO$_2$ production rates due to increased oxygen availability, as oxygenation can stimulate enzymatic activity and overall microbial growth[23]. In aquatic environments, CO$_2$ fluxes are typically controlled by diffusion and the accumulation of CO$_2$ is buffered by the carbonate system[24,25]. Streams and rivers typically show higher gas fluxes than lentic ecosystems due to higher turbulence and, thus, higher gas exchange coefficients[26].

CO$_2$ emissions from dry inland waters (mean = 186 mmol m$^{-2}$ d$^{-1}$) were in the same range, but significantly lower, than those from adjacent uphill soils which had not been previously inundated (mean ± SD = 222 ± 277 mmol m$^{-2}$ d$^{-1}$, median = 144, n = 196) (Wilcoxon signed rank test, $P < 0.05$) (Supplementary Fig. 1). Previously inundated sediments and terrestrial (uphill) soils are distinct environments in terms of their physical structure, biogeochemical dynamics, and biological communities[16,27,28]. Therefore, one plausible explanation for the observed difference in CO$_2$ emissions is the possible potential for higher root respiration in soils compared with desiccated sediments. Root respiration typically accounts for 50% of total soil respiration but may reach up to 90%[29,30]. Furthermore, organic matter content, which would fuel CO$_2$ production, was greater in uphill soils (mean ± SD = 8 ± 8%) than in dry inland waters (mean ± SD = 6 ± 7%) (Kruskal–Wallis Test, $P < 0.001$).

We observed CO$_2$ uptake by the exposed sediments at eight sites (4% of total) and by the uphill soils at five sites (3% of total). In soils, a net uptake of atmospheric CO$_2$ has been related to the dissolution of CO$_2$ in pore water and carbonate weathering[31], but direct evidence from dry inland waters supporting these mechanisms is currently missing[12].

**Homogeneity among climate zones and ecosystem types.** Our global study did not reveal significant differences in CO$_2$ fluxes between climate zones (Fig. 2). Nonetheless, this result needs to be interpreted with caution due to the unbalanced sampling sizes and the underrepresentation of sites in the polar zone. CO$_2$ emissions from polar (mean ± SD = 60 ± 58 mmol m$^{-2}$ d$^{-1}$, median = 36), continental (mean ± SD = 174 ± 140 mmol m$^{-2}$ d$^{-1}$, median =

125), temperate (mean ± SD = 178 ± 308 mmol m$^{-2}$ d$^{-1}$, median = 99), arid (mean ± SD = 233 ± 470 mmol m$^{-2}$ d$^{-1}$, median = 61) and tropical sites (mean ± SD = 236 ± 403 mmol m$^{-2}$ d$^{-1}$, median = 69) all fell within the same range (Fig. 2). CO$_2$ emissions from temperate sites experiencing dry winters (16% of temperate sites) were significantly lower than emissions from temperate sites located in either dry-summer locations (13%) or those lacking dry seasons (71%) (Kruskal–Wallis Test, $P < 0.05$). This result indicates an effect of the interaction between temperature and moisture with hot and wet conditions facilitating high gas fluxes.

All studied lentic ecosystem types (i.e. reservoirs, lakes and ponds) showed higher CO$_2$ emissions from dry sediments than globally estimated for their inundated stages (Fig. 2). CO$_2$ emissions from dry sediments of ponds (mean ± SD = 267 ± 221 mmol m$^{-2}$ d$^{-1}$, median = 252) were significantly higher than those from streams (mean ± SD = 128 ± 218 mmol m$^{-2}$ d$^{-1}$, median = 64) and reservoirs (mean ± SD = 194 ± 478 mmol m$^{-2}$ d$^{-1}$, median = 82) (Kruskal–Wallis Test, $P < 0.05$) and marginally higher than those from lakes (mean ± SD = 215 ± 353 mmol m$^{-2}$ d$^{-1}$, median = 111) (Fig. 2). This result emphasises the global importance of small waterbodies[17,18,21,32], which are extremely prevalent global biogeochemical hotspots[21,33], and which furthermore frequently exist as only temporary ecosystems, increasing the proportional relevance of their dry fluxes[17]. Possible reasons for higher CO$_2$ emissions from dry ponds compared with other ecosystem types may be high temperature and a large perimeter to area ratio which leads to organic matter accumulation in their sediments. Indeed, higher CO$_2$ emissions from ponds match the higher content of organic material we found at desiccated pond sites (18 ± 20%) compared with streams (3 ± 4%, Kruskal–Wallis Test, $P < 0.05$), lakes (14 ± 17%), and reservoirs (10 ± 11%).

Variation in CO$_2$ fluxes from dry inland waters was higher between sites than between climate zones or between the studied ecosystem types (Fig. 2). Hence, local conditions prevailed over geographical patterns, indicating that the drivers of CO$_2$ emissions in dry inland waters might be universal, thus facilitating the evaluation of this process at the global scale.

**Drivers of CO$_2$ emissions from dry inland waters.** The relationships between CO$_2$ fluxes and environmental variables were modelled using a linear mixed-effects model (LMM) (Fig. 3). LMM modelling of CO$_2$ fluxes explained 39% of the total variance by the fixed effects and 52% by the entire model (Supplementary

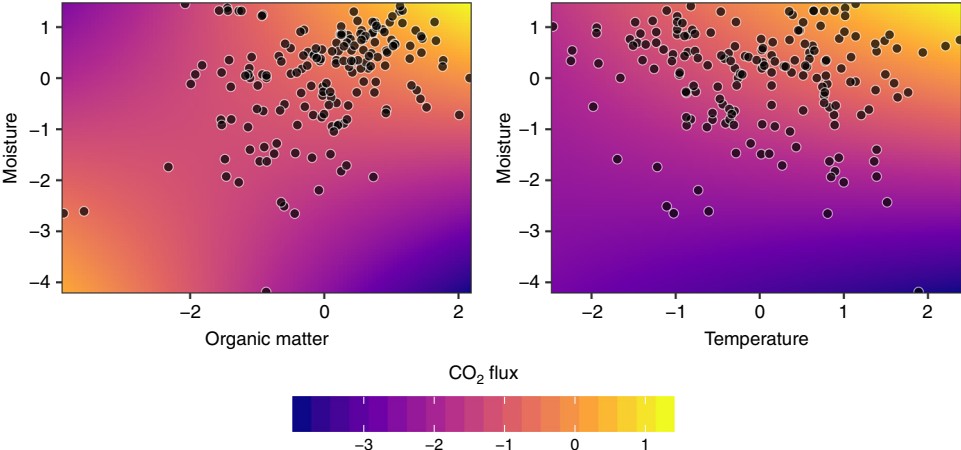

**Fig. 3 Response of CO$_2$ fluxes to environmental variables.** Left, moisture against organic matter. Right, moisture against temperature. Original values of moisture (%), organic matter (%) and CO$_2$ flux (mmol m$^{-2}$ d$^{-1}$) are shown in a log$_{10}$-transformed and z-transformed scale. Original values of temperature (°C) are shown in a z-transformed scale. Relationships arise from the linear mixed-effects model analysis.

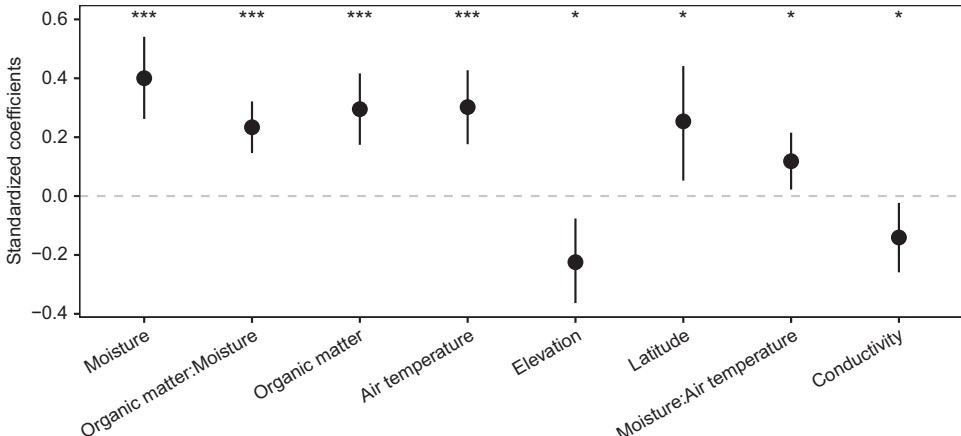

**Fig. 4 Resulting coefficients from the linear mixed-effects model.** Error bars indicate 95% confidence interval. Variables are shown in decreasing order of significance (analysis of variance, \*\*\*P < 0.001, \*P < 0.05). Moisture, elevation and conductivity have been log$_{10}$-transformed and all variables have been z-transformed prior to analysis. Colons indicate interaction between the respective variables.

Table 2). Organic matter content, moisture, temperature and the interaction between organic matter content and moisture were the strongest predictors of CO$_2$ fluxes from dry inland waters (analysis of variance, P < 0.001; Fig. 3, Supplementary Table 2), followed by the interaction of temperature with moisture and elevation, latitude and conductivity (analysis of variance, P < 0.05; Fig. 4). These results indicate that there is a universal control mechanism across ecosystems and climates. Under low-moisture conditions, neither the organic matter content of the sediments nor their temperature affected CO$_2$ emissions, because microbial activity is inhibited by water limitation[34] (Fig. 3, Supplementary Table 2). Hence, an increase in organic matter or temperature alone is not enough to produce high CO$_2$ emissions. In contrast, high moisture facilitates the contact between microorganisms and available labile organic matter, but high moisture in combination with limited availability of organic matter to fuel CO$_2$ production results in low CO$_2$ emissions (Fig. 3). The same effect can be observed when low temperature limits microbial activity. Beyond the joint influence of moisture and organic matter on CO$_2$ emissions induced by respiration, abiotic processes depending on pore water characteristics can affect the C cycle of drying sediments[35]. Abiotic CO$_2$ emissions linked to carbonate precipitation and dissolution can be a potent source of total C emissions[36]. Sediment pore water can additionally lead to an uncoupling of

CO$_2$ production and emissions in dry sediments due to reduced physical gas transfer rates[26].

Elevation, latitude and conductivity likely represent local geographical conditions as well as small-scale patterns, which were not included in our sampling design. These could be, for instance, organic matter quality/lability, the presence of terrestrial vegetation (primary production), CO$_2$ inputs via groundwater discharge, composition of the microbial community, or carbonate formation, which previous studies have identified as being potentially important[16,17]. Finally, antecedent conditions such as the time since desiccation or the past input of organic matter into the system may also influence CO$_2$ emissions[37,38].

## Discussion

Our study encompasses 196 dry inland waters (and adjacent uphill terrestrial sites), spanning all major lotic and lentic aquatic ecosystem types and global climate zones. We show that drivers of CO$_2$ emissions from desiccated sediments to the atmosphere are globally consistent, and are better predictors of CO$_2$ emissions compared with regional variability associated with climate and ecosystem type. CO$_2$ emissions from dry inland waters were generally lower than those reported for flowing streams and rivers[11], but higher than from lentic waters[11,20,21]. This pattern is consistent for most ecosystems across all climate zones. These

results strongly indicate that dry inland waters are significant and globally prevalent sources of $CO_2$ to the atmosphere[12].

Desiccated areas are usually excluded from global inventories of water bodies[39] and so their contribution is missing in current global C budgets of inland waters[11,14,20]. A global upscaling of our measured $CO_2$ emissions results in global C emissions from dry inland waters of $0.12 \pm 0.13$ Pg C y$^{-1}$ (Supplementary Table 3), which is equivalent to $6 \pm 6\%$ of the currently estimated global C emissions from inland waters (2.1, range = 1.56 – 2.94 Pg C y$^{-1}$)[11]. Because of the considerable variation of global $CO_2$ emissions from dry inland waters, a final evaluation of their contribution to global $CO_2$ emissions from inland waters remains difficult. However, partial exposure of sediments might become disproportionally more relevant in regions with a projected increase in water stress due to global change[22,40]. Hence, $CO_2$ emissions from dry inland waters could increase significantly in more arid regions, and other climate zones subject to large seasonality such as monsoon climates, even if the increase in global emissions remains modest.

In any case, the net effect of including desiccated areas in current global inventories of C emissions from inland waters would depend on how desiccated areas have been considered in former studies, which is not always traceable. For instance, excluding $CO_2$ emissions from dry inland waters, as done in recent studies[11] would at first sight imply an underestimation of current inland waters $CO_2$ emissions to the atmosphere. However, the mistaken assignment of an intermittent stream as a permanent flow area may instead result in an overestimation of fluxes, as flowing waters appear to generally emit more $CO_2$ than the dry phases of intermittent rivers. On the contrary, dry areas of ponds, lakes and reservoirs, which global $CO_2$ flux assessments assigned wrongly as wetted areas would likely result in an underestimation of net fluxes. Recent global emission inventories have either disregarded desiccated areas[11,41] (i.e. likely underestimating emissions) or incorporated intermittent streams using rough approaches, probably underestimating their area[19,38] (i.e. likely overestimating emissions). Certainly, no current global estimate considers desiccated areas in ponds, lakes and reservoirs, and thus these fluxes are likely to be underestimated. In sum, an assessment of the impact of desiccated areas on the global inland waters C inventory requires a much more accurate estimate of temporarily and permanently desiccated areas. Recent developments in remote sensing[10] may help to incorporate desiccated areas from lakes, reservoirs and large rivers, but an accurate estimate of intermittent stream and pond area is still a challenging endeavour considering most desiccated areas in vast regions of the world are obscured by cover (e.g. dense trees, clouds). This should be a research priority if $CO_2$ emissions from stream, rivers and ponds are to be accurately incorporated into global inland water C flux estimates.

We also note that our global estimates of dry $CO_2$ emissions are likely to be conservative as the global surface area of desiccated inland waters is likely underestimated[12]. Furthermore, rewetting events are short periods of high biogeochemical activity that may contribute significantly to $CO_2$ fluxes[42] and are not purposely included in our estimates. Rapid pulses of $CO_2$ production following rewetting have been observed in a variety of soil ecosystems[42,43] as well as in dry river beds[37,38].

The substantial variation between sites demands a better understanding of the underlying mechanisms driving $CO_2$ emissions from dry inland waters to the atmosphere. Further research is necessary to determine the effect of temporal and seasonal variability on $CO_2$ emissions from dry inland waters, to link these emissions with the consumptive loss of sediment organic matter and to assess the role of growing vegetation on net $CO_2$ emissions. Furthermore, little is known about the emissions

of other GHGs such as methane ($CH_4$) or nitrous oxide ($N_2O$) from dry sediments of inland waters. While desiccation and subsequent oxygenation of the sediment might minimise emissions of $CH_4$ from dry sediments[44], there are nevertheless reports of high $CH_4$ emissions immediately after drying[3,42]. In addition, we expect desiccation to have a major impact on nitrogen cycling with consequences for $N_2O$ emissions; that is lower denitrification but higher nitrification, with both processes contributing to $N_2O$ production[45]. Further research is necessary to improve our understanding of the magnitude and drivers of the emissions of these GHGs from dry inland waters.

Upscaling $CO_2$ emissions from dry inland waters for global estimates is particularly relevant because dry areas are predicted to increase in the future due to the observed and predicted decline in inland water levels following projected trends in global climate[22,40] and human activities[10,46]. An improved understanding of the global patterns and drivers of desiccated sediment $CO_2$ emissions to the atmosphere is thus crucial for an accurate understanding of contemporary landscape C cycling, as well as predictions of future atmospheric $CO_2$ concentrations due to anthropogenic activities.

## Methods

**Sampling design.** To obtain a global data set of $CO_2$ fluxes and sediment and soil characteristics, measurements were performed by 24 teams in 17 countries. The methodology was defined in a standardised sampling protocol. The objective of this study was to record a dataset with the best possible geographical coverage. Therefore, and to enable all partners to conduct the sampling campaigns, we chose parameters and methods that were relatively easy to measure and to apply. All sites were chosen by the local teams, who ensured that sites were independent and not hydrologically connected in a direct upstream–downstream relationship. Sampling was performed at two locations on each site, the dry sediment of the water body and the adjacent uphill soil. The measurements of $CO_2$ flux and additional soil and sediment parameters were performed at three plots, typically separated by a few metres, within each site. In cases where the whole ecosystem had dried up (e.g. small ponds, ephemeral streams), measurements were performed at representative parts of the bare sediment. In case of partial drying, measurements were performed at the emerged sediments at the shore. All raw data were collected and centrally analysed. The sampling sites were classified into four inland water ecosystem types, based on the information provided by the local sampling teams. We defined a stream as a natural watercourse that flows permanently or intermittently[47], a lake as a naturally occurring low point in the landscape that contains standing water at least during certain periods[48], a reservoir as a human-made lake[48] and a pond as a standing surface water body type that is considerably smaller than a lake or reservoir[49].

**$CO_2$ flux.** Closed chamber measurements were performed to measure the $CO_2$ flux directly. Opaque chambers connected to an infra-red gas analyser were inserted about 1 cm into the sediment. The $CO_2$ concentration within the chamber was monitored for <5 min and the flux was determined by a linear regression based on the change in $CO_2$ partial pressure (p$CO_2$) over time. The $CO_2$ flux (mmol m$^{-2}$ d$^{-1}$) was calculated according to Eq. (1), where $dpCO_2/dt$ is the slope of the change in p$CO_2$ with time [μatm d$^{-1}$], $V$ is the volume of the chamber [m$^3$], $S$ is the surface area covered by the chamber [m$^2$], $T$ is the air temperature [K] and $R$ is the ideal gas constant = 8.314 l atm K$^{-1}$ mol$^{-1}$.

$$F_{CO_2} = \left(\frac{dpCO_2}{dt}\right) \cdot \left(\frac{V}{RTS}\right) \quad (1)$$

When intrusion of the chamber to the ground was prevented (e.g. by a stony surface), the chamber was sealed to the ground using clay[50]. Chamber placement was restricted to plots with bare ground and sampling of vegetated surface was avoided. Positive values represent emissions from the sediment to the atmosphere while negative values indicate an inflow from the atmosphere to the ground.

**Environmental variables.** A set of 14 environmental variables was estimated for each site. Of these, ten variables were measured in situ or determined locally. We measured air and sediment temperature, determined sediment texture following the FAO manipulative test[51] and collected sediment samples at every measurement plot. For measuring sediment temperature, the sensing head of a thermometer was inserted 2–3 cm into the sediment. In the laboratory, one part of fresh sediment sample was mixed with 2.5 parts distilled water and pH and conductivity were measured in the suspension using conventional electrodes. Furthermore, we determined water content and organic matter gravimetrically by drying 5 g of fresh sediment at 105 °C until constant weight, followed by combustion at 500 °C.

Five major climate zones were assigned to sites based on their location using the 'World Maps Of Köppen-Geiger Climate Classification' dataset[52]: tropical (Köppen–Geiger group A), arid (Köppen–Geiger group B), temperate (Köppen–Geiger group C), continental (Köppen–Geiger group D) and polar (Köppen–Geiger group E). For an in-depth analysis of temperate sites, the 2nd-order sub-groups dry-summer (Köppen–Geiger group Cs), dry-winter (Köppen–Geiger group Cw) and without-dry-seasons (Köppen–Geiger group Cf) were additionally distinguished. Annual mean temperature and annual precipitation for each site were taken from the WorldClim database[53].

**Data analysis.** We tested the influence of environmental variables (Supplementary Table 4) on $CO_2$ emissions from dry inland waters by fitting LMM to the response variable $CO_2$ flux. This was done using the function lmer of the lme4 package[54] of R[55]. We selected air temperature, organic matter content, texture, moisture, conductivity, latitude, elevation, type of ecosystem (i.e. stream, lake, reservoir, pond) pH, climate zone, annual mean temperature and annual precipitation as well as 2nd order interactions between moisture, temperature and organic matter as fixed effects. Air temperature was included instead of sediment temperature because of the high correlation between these parameters ($r = 1$). We included the team performing the analysis as a random effect to account for unmeasured team-level variation (random intercepts). Afterwards the model was simplified by removing non-significant predictors from the model (Supplementary Table 4).

For all steps of the analysis, one value per parameter was obtained per location and site by averaging the three measured plots. We log-transformed $CO_2$ flux ($x + 28$), conductivity, organic matter content, moisture ($x + 0.1$) and elevation to meet the condition of normality and homogeneity of variance. All statistical analyses were conducted using R version 3.4.4[55]. Statistical tests were considered significant at $P < 0.05$.

## Data availability
The source data underlaying Figs. 1–4, Supplementary Fig. 1 and Supplementary Tables 2–4 are provided as a Source data file.

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

## Acknowledgements

This study was made possible thanks to a large collective effort of a global research network called dryflux (www.ufz.de/dryflux). We would like to thank numerous helpers for their assistance during field work. This research was inspired by GLEON (Global Lake Ecological Observatory Network). This work was supported by the German Research Foundation (DFG, KO1911/6-1 and GR1540/23-1) to P.S.K. and H.P.G., the Spanish Ministry of Science, Innovation and Universities (C-HYDROCHANGE, CGL2017-86788-C3-3-P and CGL2017-86788-C3-2-P) to B.O. and R.M., the Spanish Government (CGL2016-77487-R), the Basque Government (IT951-16), the BBVA Foundation (06417) to D.vS. and A.E., the European Research Council (FP7/2007-2013, ERC grant agreement 336642) to A.L. and R.F.M., CNPq (310033/2017-9) to A.M.A., the Carlsberg Foundation (CF16-0325) to T.R. and A.P., the Nederlandse Organisatie voor Wetenschappelijk Onderzoek (NWO, Veni Grant 86312012) to S.K., the Estonian Ministry of Education and Research (IUT 21-02) and the Estonian Research Council grant (PUT PSG32, PUT1598) to A.L. and E-I.R., the National Research Foundation of Korea (2017R1D1A1B06035179) to J-H.P., German Federal Ministry of Education and Research (BMBF) CLIENT programme (grant: 2WCL1337A) and German Academic Exchange Service (DAAD, grant 57218695) to M.A.F., the Seneca Foundation (20645/ JLI/18) to M.M.S.M. and M.I.A. N.C. was supported by Beatriu de Pinós grant (2016-00215), A.P. by the Ramón Areces Foundation postgraduate studies programme, R.dC. by the University of Murcia (FPU R-269/2014), E.S.O.J. by the Erasmus+ Programme NON-EU 2017/2018, J.R.P. by the Coordenação de Aperfeiçoamento de Pessoal de Nível Superior (CAPES Finance Code 001). C.M-L. by the French Agency for Biodiversity (ONEMA-AFB, Action 13, 'Colmatage, échange snappe-rivière et processus biogéochimiques'), R.M. by the project C-HydroChange, funded by the Spanish Agencia Estatal de Investigación (AEI) and Fondo Europeo de Desarrollo Regional (FEDER) under the contract FEDER-MCIU-AEI/ CGL2017-86788-C3, P.S.K. by a grant for a short-term scientific mission within the COST Action CA15113 (SMIRES, Science and Management of Intermittent Rivers and Ephemeral Streams, www.smires.eu) and GLEON (student travel grant). In memory of our esteemed colleague and friend Julia Howitt, who passed away after this paper was accepted. She was full of enthusiasm for this work and will be deeply missed by her colleagues around the world.

## Author contributions

R.M., N.C., D.vS, H-P.G., B.O., and M.K. initiated the project and designed the sampling campaign. P.S.K., N.C., D.vS., H-P.G., M.K., B.O., M.A.F., N.K., N.B., J.A.H., C.M-L., A.P., G.F., R.A., T.R., M.I.A., G.O., J.R.P., A.L., RdC., A.M.A., S.C-F., S.B., J.C., R.F.M., F.R., E-I.R., T.D., F.R., A.L., U.O., J-H.P., H.W., S.K., R.G., C.F., A.E., M.M.S-M., C.M.F., M.M., E.S.O.J., C.C.M., L.G-G., C.L., Q.Z., R.M. measured $CO_2$ fluxes, sampled field data and processed this material. P.S.K. carried out the data compilation and database management. P.S.K., N.C and R.M. performed the data analyses. P.S.K. led the writing of the manuscript with notable contributions by N.C., D.vS., H-P.G., B.O., M.K. and R.M. All the other authors commented on and contributed to revising draft versions. Their order was computed randomly.

## Competing interests

The authors declare no competing interests.

## Additional information

P.S. Keller [1✉], N. Catalán [2,3], D. von Schiller [4], H.-P. Grossart [5,6], M. Koschorreck [1], B. Obrador [4], M. A. Frassl [1,7], N. Karakaya [8], N. Barros [9], J. A. Howitt [10,37], C. Mendoza-Lera [11], A. Pastor [12], G. Flaim [13], R. Aben [14], T. Riis [12], M. I. Arce [15], G. Onandia [16], J. R. Paranaíba [9], A. Linkhorst [17], R. del Campo [18,19], A. M. Amado [9,20], S. Cauvy-Frauniée [11], S. Brothers [21], J. Condon [22], R. F. Mendonça [9], F. Reverey [16], E.-I. Rõõm [23], T. Datry [11], F. Roland [9], A. Laas [23], U. Obertegger [13], J.-H. Park [24], H. Wang [25], S. Kosten [14], R. Gómez [19], C. Feijoó [26], A. Elosegi [27], M. M. Sánchez-Montoya [19], C. M. Finlayson [28,29], M. Melita [30], E. S. Oliveira Junior [14,31], C. C. Muniz [31], L. Gómez-Gener [32], C. Leigh [7,33,34,35], Q. Zhang [36] & R. Marcé [2,3]

[1]Department of Lake Research, Helmholtz Centre for Environmental Research—UFZ, Magdeburg, Germany. [2]Catalan Institute for Water Research (ICRA), Girona, Spain. [3]Universitat de Girona, Girona, Spain. [4]Department of Evolutionary Biology, Ecology and Environmental Sciences, University

of Barcelona, Barcelona, Spain. [5]Department Experimental Limnology, Leibniz Institute of Freshwater Ecology and Inland Fisheries (IGB), Neuglobsow, Germany. [6]Institute of Biology and Biochemistry, Potsdam University, Potsdam, Germany. [7]Australian Rivers Institute, Griffith University, Nathan, QLD, Australia. [8]Department of Environmental Engineering, Bolu Abant Izzet Baysal University, Bolu, Turkey. [9]Biology Department, Federal University of Juiz de Fora, Minas Gerais, Brazil. [10]School of Agricultural and Wine Sciences, Institute for Land, Water and Society, Charles Sturt University, Wagga Wagga, NSW, Australia. [11]INRAE, UR RiverLy, Centre de Lyon-Villeurbanne, Villeurbanne, France. [12]Department of Bioscience, Aarhus University, Aarhus, Denmark. [13]Department of Sustainable Agro-ecosystems and Bioresources, Research and Innovation Centre, Fondazione Edmund Mach, San Michele all'Adige, Italy. [14]Department of Aquatic Ecology and Environmental Biology, Institute for Water and Wetland Research, Radboud University, Nijmegen, the Netherlands. [15]Leibniz-Institute of Freshwater Ecology and Inland Fisheries (IGB), Berlin, Germany. [16]Leibniz Centre for Agricultural Landscape Research (ZALF), Müncheberg, Germany. [17]Department of Ecology and Genetics, Limnology, Uppsala University, Uppsala, Sweden. [18]Department of Ecology, University of Innsbruck, Innsbruck, Austria. [19]Department of Ecology and Hydrology, University of Murcia, Murcia, Spain. [20]Departamento de Oceanografia e Limnologia, Universidade Federal do Rio Grande do Norte, Natal, Brazil. [21]Department of Watershed Sciences and Ecology Center, Utah State University, Logan, UT, USA. [22]Graham Centre for Agricultural Innovation, Charles Sturt University and New South Wales Department of Primary Industries, Wagga Wagga, NSW, Australia. [23]Chair of Hydrobiology and Fishery, Institute of Agricultural and Environmental Sciences, Estonian University of Life Sciences, Tartu, Estonia. [24]Department of Environmental Science and Engineering, Ewha Womans University, Seoul, Republic of Korea. [25]State Key Laboratory of Freshwater Ecology and Biotechnology, Institute of Hydrobiology, Chinese Academy of Sciences, Wuhan, China. [26]Programa Biogeoquímica de Ecosistemas Dulceacuícolas (BED), Instituto de Ecología y Desarrollo Sustentable (INEDES, CONICET-UNLu), Luján, Argentina. [27]Department of Plant Biology and Ecology, University of the Basque Country (UPV/EHU), Bilbao, Spain. [28]Institute for Land, Water and Society, Charles Sturt University, Albury, Australia. [29]IHE Delft, Institite for Water Education, Delft, the Netherlands. [30]Water Research Institute—National Research Council (IRSA-CNR), Montelibretti (Rome), Italy. [31]Center of Etnoecology, Limnology and Biodiversity, Laboratory of Ichthyology of the Pantanal North, University of the State of Mato Grosso, Cáceres, Brazil. [32]Department of Ecology and Environmental Science, Umeå University, Umeå, Sweden. [33]Institute for Future Environments and School of Mathematical Sciences, Science and Engineering Faculty, Queensland University of Technology (QUT), Brisbane, QLD, Australia. [34]ARC Centre of Excellence for Mathematical & Statistical Frontiers (ACEMS), Brisbane, QLD, Australia. [35]Biosciences and Food Technology Discipline, School of Science, RMIT University, Bundoora, VIC, Australia. [36]Nanjing Institute of Geography & Limnology (NIGLAS), Chinese Academy of Sciences, Nanjing, China. [37]Deceased: J. A. Howitt. ✉email: philipp.keller@ufz.de

