## [Peer Review File · Nature Communications]

Reviewers' comments:

Reviewer #1 (Remarks to the Author):

Review of Keller et al. Global CO₂ emissions from dry inland water share common drivers across ecosystems. Submitted to Nature Communications.

In this manuscript, the authors use data collected from 196 sites across 6 continents to assess the contribution of dry inland waters to the global carbon budget, and arrive at the conclusion that CO₂ emissions from the exposed sediment would increase the projected C budget by 0.4 to 10%. CO₂ flux from exposed sediment was an order of magnitude higher relative to lentic waters, but was typically lower than emission rates from uphill gas fluxes. The authors additionally investigated the main driving factors for CO₂ flux from dry sediments, focusing on organic matter content, temperature, moisture, and texture. Linear mixed-effects model (LMM) and Random Forest regression (RF) were used to model the response of CO₂ fluxes to the measured environmental variables. The combination of drivers that have a significant impact CO₂ emissions are organic matter vs. moisture, and temperature vs. moisture content of the sediment. The takeaway from this study is that the drivers were globally consistent within all the climate zones but vary within the ecosystem type. This indicates that there are local factors that strongly impact the emissions of CO₂.

Overall, this study provides a timely estimate of dry inland water CO₂ fluxes that is based on a broad global sampling effort. This area of research is relatively new, as most CO₂ flux estimates have focussed on water vs. land ecosystems, and those areas that undergo intermittent wet/dry cycling have mostly been ignored. Given our changing climate, which is expected to affect water body size and depth via changes in temperature and precipitation, the question of the effect of drying on gas fluxes is very pertinent. Although the final magnitude of gas flux from these dried sites is fairly low (6% ± 6% of global CO₂ flux), we think the local impacts of these drying events is likely to be very significant, particularly in already arid and semi-arid regions. This work is therefore very valuable and provides a strong context for future work in this area to tease apart local impacts of drying on gas fluxes.

Major comments:

1) It was difficult to interpret the results of the statistics. In particular, it is not currently clear what analyses were performed to produce Fig. 4 vs. Fig. 5. In reading the data analysis section in the methods (line 316-337), the authors describe an iterative use of linear mixed models and random forest models (and then another subsequent LMM?). But the two figures seem to present a different ordering of important variables (for example, Fig. 4 suggests conductivity is the least "important" variable, while Fig. 5 suggests it is the most important?). Part of the confusion stems from a lack of familiarity on our side of the use of random forest models in particular, but a more thorough explanation of why these statistical analyses were used, how to interpret them, and how to interpret seemingly contradictory results, would be very beneficial to the reader.

A few specific suggestions that might aid in this interpretation:

a) Provide a model output for both the Linear mixed-effects model (LMM) and the Random Forest (RF) in the supplementary materials.

b) For Fig. 4, it was not clear what the positive and negative values indicate for the marginal response. Do larger positive and negative values represent importance? Perhaps more detail to the statistical analysis is needed.

c) The top model set (Fig. 5) does not convey the importance or significance of the variables clearly. What does the positioning of the standard coefficients indicate?

2) The primary conclusions reported in this paper focus on the % contribution of dry flux to global inland water flux, but the value derived in this paper (6%) is underwhelming. We think, however, that this dry flux is going to be disproportionately relevant in different regions, particularly in semi-arid and arid regions that might experience more extreme changes in sediment exposure with climate change. The impact of this paper could be improved by noting some of these regions and emphasising that the local impacts could be significant, even if the global impacts are modest.

Minor Comments

1) Line 89. You state that seasonal desiccation reduces inland water surface area by ~800,000 km².

How does this compare to total SA (indicate % change?)?

2) Lines 140-142. You report a range for lentic water surface emissions, but one value for lotic. Is this a mean value? Please be consistent in what is reported here.

3) Figure 3. How were the water bodies defined, in particular reservoir vs. lake, vs. pond?

4) Line 158-159. Root respiration was mentioned in the results. How was root respiration measured or taken account of in the calculations or modelling?

5) Line 291. Greenhouse gas chambers were inserted 1 cm into the sediment for in situ measurements, but most other studies state that chambers should either be inserted with a pre-installed collar or 5 to 10 cm into the sediment to prevent leakage (e.g. Hutchinson and Livingston 2001. Vents and seals in non-steady-state chambers used for measuring gas exchange between soil and the atmosphere *European Journal of Soil Science* 52(4), 675-682. <https://dx.doi.org/10.1046/j.1365-2389.2001.00415.x>).

6) Line 229-235. This section of the discussion is a little confusing. In particular, the discussion of the influence of carbonate precipitation seems out of place, and isn't otherwise considered in most parts of the paper. Please rephrase this section for improved clarity.

7) Both air and sediment temperatures were measured. Only air temperature was used for CO₂ flux calculations, why were sediment temperatures omitted?

8) Final paragraph, p13 (lines 269-279). Would future studies include methane emissions from the desiccated water bodies? Would the variables of the methane emissions follow the same patterns as CO₂?

9) Line 179. Please explain/ define dry winter. Is this no snow? Or when the dry season occurs in winter months?

10) Desiccate is spelled differently throughout the text (desiccate vs. dessicate). (lines 81, 83, 100, 196).

11) Line 310. "5 gr" should be "5 g".

Reviewer #2 (Remarks to the Author):

This manuscript presents the results from a large-scale, collaborative study designed to assess the relative importance of CO₂ emissions from inland waters during dry conditions. Net carbon dioxide emission rates from dry aquatic sediments and adjacent upland soils were measured using chambers and IRGAs at 196 sites in different ecosystem types and climatic zones. These data were complemented with measurements of proximal and distal correlates (i.e. potential drivers) of CO₂ emissions. The authors provide a very interesting analysis and the paper is in relatively good shape. However, I identified some concerns with the manuscript, including somewhat careless preparation. I have both general and specific suggestions for its further improvement.

General Comments

Maybe the abbreviated format makes it challenging (and perhaps Fig. 1 suffices), but the introduction is more-or-less devoid of specific hypotheses or predictions.

Although the authors are to be congratulated for obtaining data from 196 sites representing a great deal of diversity in environmental conditions, they should also acknowledge the rather uneven geographic distribution of their sites – most are in Europe, with only a single site on the African continent.

Although it will necessarily be speculative, the authors should acknowledge the potential production of other greenhouse gases (e.g., methane, nitrous oxide) from desiccated aquatic sediments and state that these also deserve study.

The methods contain no information about how the random forest regression was performed. At the

very least, the authors need to provide the R package used (if they used R to do the analysis). Either way, more details are needed.

Specific Suggestions

- 1) Line 84: Replace "by" with "or from"
- 2) Line 93: Delete "between"
- 3) Line 148: "diffusion, while production of CO₂..."
- 4) Line 176 and Figs. 2 and 3: What is a "continental" climate zone? On lines 113-114 the authors state that the climate zones are "tropical, arid, temperate, boreal, and polar" but in the results "boreal" seems to have been replaced by "continental." Do these mean the same thing? This obviously needs to be cleaned up.
- 5) Line 200: Insert comma after "patterns"
- 6) Line 201: If the requisite data on drivers are available.
- 7) Line 207: "...as a result of..."
- 8) Line 216: "...(step 2). We then used..."
- 9) Line 223: "low-moisture"
- 10) Line 224: "nor their temperature"
- 11) Line 229: Delete "analogously"
- 12) Line 235: It might be good also to acknowledge the importance of antecedent conditions in driving emission rates.
- 13) Line 238: "small-scale" and insert comma after "patterns"
- 14) Line 256: Insert comma after parenthesis
- 15) Line 263: Replace "which" with "that"
- 16) Line 269: Delete "for"
- 17) Line 272: Insert comma after "matter"
- 18) Line 278: Insert comma after "cycling"
- 19) Line 310: "5 g" and delete "a"
- 20) Line 320: Define "LOI"
- 21) Supplementary Fig. 2: Typo in x-axis legend of bottom-left plot.

Reviewer #3 (Remarks to the Author):

This manuscript reports a very nice datasets of 196 CO₂ surface-to-atmosphere fluxes measured from "dry" aquatic ecosystems – streams, reservoirs, lakes, or ponds that were either permanently or temporarily lacking surface water inundation. They find that CO₂ flux did not vary by climate zone but did vary by ecosystem type; CO₂ fluxes from dry streams and reservoirs were significantly lower than CO₂ fluxes from dry ponds. As we start to understand more about how aquatic ecosystems respond to drying under a changing climate this results is particularly important. They also compare these to measured adjacent hillslope fluxes; they do not present an analysis by region or ecosystem type, but find on average across the study that hillslope fluxes were higher than dry aquatic fluxes. Finally they use literature values to compare global average fluxes of aquatic ecosystems.

Some description and details are lacking that make evaluating the methods difficult. For example, How were field sites chosen? Within field sites, how were replicate sampling locations chosen and how (was) that error propagated through the statistical analysis? How many sites were permanently dry versus temporarily, and any indication of how long they had been dry? Were these locations that typically dry seasonally or did they more recently dry? How were ecosystem types determined (ie a lake and pond are normally differentiated based on water depth, but for dry sites how was that distinction made)? Was the "whole" lake, pond, reservoir dry or were measurements made on dry margins? Which season were measurements taken and how was that incorporated into statistical analyses?

The statistical analysis design was also difficult to follow – rather than using an approach that integrated the hierarchical nature of their predictor variables (proximal and distal) (perhaps something like structural equation modeling) they used three separate statistical models – LMM with proximal variables, followed by RF, followed by a second LMM, which seemed unnecessarily cumbersome. It is also unclear whether this approach was used for $n = 196$ dry aquatic ecosystem fluxes and also $n = 196$ adjacent uphill terrestrial fluxes – and if not, why? Finally, the only reporting of model results is in two figures in the main text – the authors used AICc selection and averaging but there is no reporting of fit models in the main text or supplementary information.

Finally, I found the argument about dry inland waters being included in global budgets difficult to follow and if I understand the authors correctly I disagree with it. The authors argue that current budgets likely underestimate CO₂ fluxes because they do not include dry inland waters – based on their data and how global budgets are produced it seems the opposite may be true. The authors are right – there is not a box for dry waters in global budgets and it is probably important to include one in global budgets especially with increased drying in the future in some regions; but at the scale and resolution of global budgets those dry water areas are included – either as terrestrial or aquatic ecosystems. Based on the comparison the authors present with uphill terrestrial fluxes (higher than dry water fluxes), global budgets that consider dry water land area as terrestrial are actually overestimating fluxes. Based on the comparison with inundated aquatic ecosystems (global averages), for streams again global budgets are overestimating fluxes. For lentic waters, global averages were quite similar to measured dry fluxes (if you squint maybe you could argue for an underestimation of global fluxes for ponds). I understand the desire to place this dataset in a global context but I think there is important nuance that needs to be considered here, and would actually make the placement of the findings into a global context much richer.

Additional comments:

Ln 183 – 185 – Typo in this sentence, perhaps missing “or”.

Ln 193 – Typo in this sentence. Delete “between”?

Ln 109 – 111 – This is a prediction and not a hypothesis.

Ln 107 – Permanently? At what point does a permanently dry area become terrestrial?

Figure 1 – This simple conceptual model is not that useful – it uses a lot of space to basically list 14 variables without much nuance about how those variables are related to one another. Also, colors are not readable by somebody that is colorblind.

Ln 152 – The reporting of statistical results is confusing here – was the significance tested of the mean (lower) or variability (more variable) or both? I am also curious why the authors did not compare by ecosystem type or climate region. Also, this was a paired study – was that taken into account in statistical analysis?

Ln 173 – 179 – If I am reading correctly there are no significant results in Figure 3a; yet the authors report pattern in that data in the following paragraph (polar regions lower, etc). This seems inappropriate.

Ln 179-181 – This is the only mention of the importance of season. How season was incorporated into experimental design is not included anywhere else in the manuscript. This is really important for gas fluxes.

Ln 250 – 251 – That patterns by ecosystem type were consistent across climate zones is very interesting – those results were not presented in the text.

Ln 251 – The authors big take away in this paragraph is that dry aquatic fluxes are “a significant and globally prevalent source of CO₂ to the atmosphere” – although I find this paper quite interesting I’m not sure I agree. Line 255-256 the SD overlaps zero.

Ln 335 – How was CO₂ flux log transformed if there were negative fluxes?

In streams groundwater is an important source of CO₂ – in dry streambeds was groundwater discharge considered at all, or was in situ production considered dominant and why?

NCOMMS-19-6332178-T – Reply Letter

Reviewer #1 (Remarks to the Author):

Review of Keller et al. Global CO₂ emissions from dry inland water share common drivers across ecosystems. Submitted to Nature Communications.

In this manuscript, the authors use data collected from 196 sites across 6 continents to assess the contribution of dry inland waters to the global carbon budget, and arrive at the conclusion that CO₂ emissions from the exposed sediment would increase the projected C budget by 0.4 to 10%. CO₂ flux from exposed sediment was an order of magnitude higher relative to lentic waters, but was typically lower than emission rates from uphill gas fluxes. The authors additionally investigated the main driving factors for CO₂ flux from dry sediments, focusing on organic matter content, temperature, moisture, and texture. Linear mixed-effects model (LMM) and Random Forest regression (RF) were used to model the response of CO₂ fluxes to the measured environmental variables. The combination of drivers that have a significant impact CO₂ emissions are organic matter vs. moisture, and temperature vs. moisture content of the sediment. The takeaway from this study is that the drivers were globally

consistent within all the climate zones but vary within the ecosystem type. This indicates that there are local factors that strongly impact the emissions of CO₂.

Overall, this study provides a timely estimate of dry inland water CO₂ fluxes that is based on a broad global sampling effort. This area of research is relatively new, as most CO₂ flux estimates have focussed on water vs. land ecosystems, and those areas that undergo intermittent wet/dry cycling have mostly been ignored. Given our changing climate, which is expected to affect water body size and depth via changes in temperature and precipitation, the question of the effect of drying on gas fluxes is very pertinent. Although the final magnitude of gas flux from these dried sites is fairly low (6% ± 6% of global CO₂ flux), we think the local impacts of these drying events is likely to be very significant, particularly in already arid and semi-arid regions. This work is therefore very valuable and provides a strong context for future work in this area to tease apart local impacts of drying on gas fluxes.

We are very thankful with the reviewer for the constructive and useful comments on our manuscript.

Major comments:

1) It was difficult to interpret the results of the statistics. In particular, it is not currently clear what analyses were performed to produce Fig. 4 vs. Fig. 5. In reading the data analysis section in the methods (line 316-337), the authors describe an iterative use of linear mixed models and random forest models (and then another subsequent LMM?). But the two figures seem to present a different ordering of important variables (for example, Fig. 4 suggests conductivity is the least “important” variable, while Fig. 5 suggests it is the most important?). Part of the confusion stems from a lack of familiarity on our side of the use of random forest models in particular, but a more thorough explanation of why these statistical analyses were used, how to interpret them, and how to interpret seemingly contradictory results, would be very beneficial to the reader.

We agree with the reviewer that the statistical analysis was hard to follow and unnecessarily cumbersome. We therefore completely revised and simplified the statistical approach. We initially chose this 3-step approach to avoid model overfitting caused by the large number of predictors and their interactions in relation to our sampling size. By including ecologically meaningful and relevant interactions *a-priori*, we have overcome this problem from the first step on. We therefore removed the variable selection by Random Forests and based the results on the Linear mixed-effects model (LMM). We restructured the results sections (l.211ff) accordingly: “The relationships

between CO₂ fluxes and environmental variables were modelled using a linear mixed-effects model (LMM).[...]. The former Figures 4 and 5 (Fig 3 and Fig 4 in the present version) are now only produced by using LMM. The main results and conclusions were not affected by this simplification. L. 214: “Organic matter content, moisture, temperature and the interaction between organic matter content and moisture were the strongest predictors of CO₂ fluxes from dry inland waters (analysis of variance, $p < 0.001$; Fig. 3, Supplementary Table 2), followed by the interaction of temperature with moisture and elevation, latitude, and conductivity (analysis of variance, $p < 0.05$; Fig. 4.)” which is in line with the combined results of the iterative use of LMM in the original version of the manuscript.

The description of the iterative use of LMM and RandomForest was consequently removed from the methods section (l.356): “We tested the influence of environmental variables (Supplementary Table 4) on CO₂ emissions from dry inland waters by fitting linear mixed-effects models (LMM) to the response variable CO₂ flux.[...]”

A few specific suggestions that might aid in this interpretation:

a) Provide a model output for both the Linear mixed-effects model (LMM) and the Random Forest (RF) in the supplementary materials.

We removed the Random Forest approach. Detailed results from the LMM can be found in Supplementary Table 2.

b) For Fig. 4, it was not clear what the positive and negative values indicate for the marginal response. Do larger positive and negative values represent importance? Perhaps more detail to the statistical analysis is needed.

Figure 4 is Fig. 3 in the present version. The panel b in the former figure has been removed as it was derived from the Random Forest approach, which is not used any more. Additionally, we have clarified the caption of the figure to provide a better link to the statistical analysis.

c) The top model set (Fig. 5) does not convey the importance or significance of the variables clearly. What does the positioning of the standard coefficients indicate?

We improved Fig. 5, which is Fig. 4 in the present version, by clearly indicating the ordering and the significance of the variables. Variables are decreasingly ordered according to their p value (analysis of variance).

2) The primary conclusions reported in this paper focus on the % contribution of dry flux to global inland water flux, but the value derived in this paper (6%) is underwhelming. We think, however, that this dry flux is going to be disproportionately relevant in different regions, particularly in semi-arid and arid regions that might experience more extreme changes in sediment exposure with climate change. The impact of this paper could be improved by noting some of these regions and emphasising that the local impacts could be significant, even if the global impacts are modest.

We highly appreciate this suggestion by the reviewer. We added a paragraph to the Discussion (l. 255ff) that reads: “Because of the considerable variation of global CO₂ emissions from dry inland waters, a final evaluation of their contribution to global CO₂ emissions from inland waters remains difficult. However, partial exposure of sediments might become disproportionately more relevant in regions with a projected increase in water stress due to global change^{22,41}. Hence, CO₂ emissions from dry inland waters could increase significantly in more arid regions, and other climate zones subject to large seasonality such as monsoon climates, even if the increase in global emissions remains modest.”

Minor Comments

1) Line 89. You state that seasonal desiccation reduces inland water surface area by ~800,000 km². How does this compare to total SA (indicate % change)?

We added the percentage value and improved the wording of this sentence: (l. 92). “Furthermore, seasonal desiccation affects 18% (~800,000 km²) of the global surface area covered by inland waters, exposing previously submerged sediments to the atmosphere.”

2) Lines 140-142. You report a range for lentic water surface emissions, but one value for lotic. Is this a mean value? Please be consistent in what is reported here.

We changed this and report a mean value for lentic waters in order to report one value for both types (l. 142). It now reads “Measured CO₂ emissions from dry inland waters to the atmosphere were an order of magnitude higher than average water surface emissions (water-to-atmosphere) previously reported for lentic waters (27 mmol m⁻² day⁻¹), but lower than average emissions reported for lotic waters (663 mmol m⁻² day⁻¹).”

3) Figure 3. How were the water bodies defined, in particular reservoir vs. lake, vs. pond?

We extended the method section to clarify the definition (l. 321ff). It reads: “The sampling sites were classified into four inland water ecosystem types, based on the information provided by the local sampling teams. We defined “stream” as a natural watercourse that flows permanently or intermittently⁴⁸, “lake” as a naturally occurring low point in the landscape that contains standing water at least during certain periods⁴⁹, “reservoir” as a human made lake⁴⁹ and “pond” as a standing surface water body type that is considerably smaller than a lake or reservoir⁵⁰”.

The differentiation especially between ponds and lakes is indeed a challenging subject. To our knowledge, there is no consistent definition existing of what a pond is and how ponds functionally and structurally differ from lakes. In fact, there is right now an ongoing GLEON project to create a unified definition of ponds (<https://gleon.org/research/projects/what-pond>). We defined Pond based on its size, but the classification was up to the local teams. The only alternative would have been to merge the groups lake and pond, but this would clearly have led to loss of valuable information because ponds have proven to be of particular interest not only in this study but also in the literature.

4) Line 158-159. Root respiration was mentioned in the results. How was root respiration measured or taken account of in the calculations or modelling?

Root respiration was neither measured nor considered in our calculations. However, we do consider it in the interpretation of our results. Thus, the intention of this sentence is to discuss a potential reason for the significant difference in CO₂ emission between dry sediments and soils. To make that clearer we rephrased the sentence (l. 159-160) into: “Therefore, one plausible explanation for the observed difference in CO₂ emissions is the possible potential for higher root respiration in soils than sediments. Root respiration typically accounts for 50% of total soil respiration but may reach up to 90%^{29,30}.”

5) Line 291. Greenhouse gas chambers were inserted 1 cm into the sediment for in situ measurements, but most other studies state that chambers should either be inserted with a pre-installed collar or 5 to 10 cm into the sediment to prevent leakage (e.g. Hutchinson and Livingston 2001. Vents and seals in non-steady-state chambers used for measuring gas exchange between soil and the atmosphere European Journal of Soil Science 52(4), 675-682.

<https://dx.doi.org/10.1046/j.1365-2389.2001.00415.x>).

We agree with the reviewer that the chamber placement is an important issue that must be considered carefully. However, a permanently (pre-)installed collar as it is often used in soil science cannot be used when working on dry sediments because a) it may influence hydrological flow and b) might cause sediment erosion or increase sedimentation during the flooded phase. Furthermore, the aim of achieving the greatest possible geographical coverage posed logistic challenges. For example, the use of pre-installed collars was not possible at a number of remote sites that were only visited once by the teams for our sampling campaign. Requiring pre-installed experimental set-up

would have meant waiving these sites, and we decided that the inclusion of all possible sites was more important for this study. Our experience with gas flux measurements on dry inland waters does not indicate a very strong perturbation on CO₂ emissions due to the lack of a pre-installed collar.

6) Line 229-235. This section of the discussion is a little confusing. In particular, the discussion of the influence of carbonate precipitation seems out of place, and isn't otherwise considered in most parts of the paper. Please rephrase this section for improved clarity.

Abiotic processes such as carbonate precipitation have a strong influence on C cycling and, thus, CO₂ emissions. Thus, we consider that they need to be discussed here as an emission's driver directly influenced by sediment moisture. We rephrased the section for more clarity (l. 225ff): "Beyond the joint influence of moisture and organic matter on CO₂ emissions induced by respiration, abiotic processes depending on pore water characteristics can affect the C cycle of drying sediments³⁵. Abiotic CO₂ emissions linked to carbonate precipitation and dissolution can be a potent source of total C emissions³⁶. Sediment pore water can additionally lead to an uncoupling of CO₂ production and emissions in dry sediments due to reduced physical gas transfer rates²⁶."

7) Both air and sediment temperatures were measured. Only air temperature was used for CO₂ flux calculations, why were sediment temperatures omitted?

Values of sediment and air temperature were highly collinear ($r = 1$) providing the same statistical information. We decided to remove one of them to reduce the number of predictors. We omitted sediment temperature because a small number of outliers led to better model fits using air temperature instead. We added this information to the methods section (l. 362-363)

8) Final paragraph, p13 (lines 269-279). Would future studies include methane emissions from the desiccated water bodies? Would the variables of the methane emissions follow the same patterns as CO₂?

We agree with the reviewer that emissions of other GHGs like methane and nitrous oxide from dry sediments are of high importance and deserve to be studied more intensely. We therefore included this paragraph in the discussion (l. 293ff): "Furthermore, little is known about the emissions of other GHGs such as methane (CH₄) or nitrous oxide (N₂O) from dry sediments of inland waters. While desiccation and subsequent oxygenation of the sediment might minimize emissions of CH₄ from dry sediments⁴⁵, there are nevertheless reports of high CH₄ emissions immediately after drying^{3,43}. Additionally, we expect desiccation to have a major impact on nitrogen cycling with consequences for N₂O emissions; that is lower denitrification but higher nitrification, with both processes contributing to N₂O production⁴⁶. Further research is necessary to improve our understanding of the magnitude and drivers of the emissions of these GHGs from dry inland waters."

9) Line 179. Please explain/ define dry winter. Is this no snow? Or when the dry season occurs in winter months?

All terminology regarding climate zones refers to the Köppen-Geiger classification. We extended the methods section for more clarity (l. 349ff). It now reads: "Five major climate zones were assigned to sites based on their location using the "World Maps Of Köppen-Geiger Climate Classification" data set²²: tropical (Köppen-Geiger group A), arid (Köppen-Geiger group B), temperate (Köppen-Geiger group C), continental (Köppen-Geiger group D) and polar (Köppen-Geiger group E). For an in-depth analysis of temperate sites, the 2nd order sub-groups dry-summer (Köppen-Geiger group Cs), dry-winter (Köppen-Geiger group Cw) and "without dry seasons" (Köppen-Geiger group Cf) were additionally distinguished."

Temperate climates with dry winter are defined as:

$$P_{wmin} < P_{smin} \text{ and } P_{smax} > 10 P_{wmin}$$

Where P_{wmin} , P_{smin} and P_{smax} are defined as the lowest and highest monthly precipitation values for the summer and winter half-years on the hemisphere considered. We have decided not to explicitly state the underlying definitions of those groups in order to keep this section as concise as possible. They can be found in the cited literature (DOI: 10.1127/0941-2948/2006/0130.).

10) Desiccate is spelled differently throughout the text (desiccate vs. dessicate). (lines 81, 83, 100, 196).

We changed all occurrences of „dessicate“ into „desiccate“

11) Line 310. “5 gr” should be “5 g”.

Done

Reviewer #2 (Remarks to the Author):

This manuscript presents the results from a large-scale, collaborative study designed to assess the relative importance of CO₂ emissions from inland waters during dry conditions. Net carbon dioxide emission rates from dry aquatic sediments and adjacent upland soils were measured using chambers and IRGAs at 196 sites in different ecosystem types and climatic zones. These data were complemented with measurements of proximal and distal correlates (i.e. potential drivers) of CO₂ emissions. The authors provide a very interesting analysis and the paper is in relatively good shape. However, I identified some concerns with the manuscript, including somewhat careless preparation. I have both general and specific suggestions for its further improvement.

We are very grateful for the reviewer comments and time spent on our manuscript.

General Comments

Maybe the abbreviated format makes it challenging (and perhaps Fig. 1 suffices), but the introduction is more-or-less devoid of specific hypotheses or predictions.

The reviewer is right that our hypotheses were not well defined. We restructured the introduction to be more precise and removed Fig. 1, which we believe did not suffice to clarify our hypothesis. The essential section now reads (l. 112ff): “We hypothesize that CO₂ emissions from dry inland waters are above average compared to reported inland aquatic rates, thus making emissions from dry inland waters globally relevant. We further hypothesize that sediment-atmosphere emissions vary as a function of parameters controlling CO₂ production rates (such as organic matter supply, temperature, and moisture) and parameters controlling the transport of gas to the atmosphere (e.g., sediment texture) as well as geographical properties of the sampling locations which influence the biogeochemical conditions.”

Although the authors are to be congratulated for obtaining data from 196 sites representing a great deal of diversity in environmental conditions, they should also acknowledge the rather uneven geographic distribution of their sites – most are in Europe, with only a single site on the African continent.

We agree with the reviewer that the sampling sites of the study are unevenly distributed. We acknowledged this by adding a sentence at the beginning of the results section (l. 139): “The sampled sites include a great diversity of environmental conditions (Fig. 1), although the collaborative nature of the study precluded an even geographical distribution of sampling efforts, and sites in the temperate zone dominate the dataset.” In fact, our dataset includes more than one single site on the African continent. We added a displacement to the symbols in the map (former

Figure 2, now Figure 1) to improve the visualisation.

Although it will necessarily be speculative, the authors should acknowledge the potential production of other greenhouse gases (e.g., methane, nitrous oxide) from desiccated aquatic sediments and state that these also deserve study.

We agree with both reviewers that other greenhouse gases definitely deserve study. We therefore included a paragraph to the discussion acknowledging their importance (l. 293ff) : “Furthermore, little is known about the emissions of other GHGs such as methane (CH₄) or nitrous oxide (N₂O) from dry sediments of inland waters. While desiccation and subsequent oxygenation of the sediment might minimize emissions of CH₄ from dry sediments⁴⁵, there are nevertheless reports of high CH₄ emissions immediately after drying^{3,43}. Additionally, we expect desiccation to have a major impact on nitrogen cycling with consequences for N₂O emissions; that is lower denitrification but higher nitrification, with both processes contributing to N₂O production⁴⁶. Further research is necessary to improve our understanding of the magnitude and drivers of the emissions of these GHGs from dry inland waters.”

The methods contain no information about how the random forest regression was performed. At the very least, the authors need to provide the R package used (if they used R to do the analysis). Either way, more details are needed.

We revised the statistical approach and no longer use Random Forests regressions. We therefore removed all details on RandomForest from the manuscript.

Specific Suggestions

1) Line 84: Replace “by” with “or from”

Done

2) Line 93: Delete “between”

Done

3) Line 148: “diffusion, while production of CO₂...”

Done

4) Line 176 and Figs. 2 and 3: What is a “continental” climate zone? On lines 113-114 the authors state that the climate zones are “tropical, arid, temperate, boreal, and polar” but in the results “boreal” seems to have been replaced by “continental.” Do these mean the same thing? This obviously needs to be cleaned up.

We appreciate the reviewer’s careful reading of the manuscript. The terms “boreal” and “continental” both refer to the Köppen-Geiger group “D”. The revised manuscript now only uses the phrase “continental”. We added more details to the methods section to be more specific about the climate zones (l. 349ff): “Five major climate zones were assigned to sites based on their location using the "World Maps Of Köppen-Geiger Climate Classification" data set²²: tropical (Köppen-Geiger group A), arid (Köppen-Geiger group B), temperate (Köppen-Geiger group C), continental (Köppen-Geiger group D) and polar (Köppen-Geiger group E).”

5) Line 200: Insert comma after “patterns”

Done

6) Line 201: If the requisite data on drivers are available.

Done

7) Line 207: “...as a result of...”

Done

8) Line 216: "...(step 2). We then used..."

Done

9) Line 223: "low-moisture"

Done

10) Line 224: "nor their temperature"

Done

11) Line 229: Delete "analogously"

Done

12) Line 235: It might be good also to acknowledge the importance of antecedent conditions in driving emission rates.

We agree with the reviewer and acknowledge in the present version the importance of antecedent conditions in the sentence (l. 235ff): "Finally, antecedent conditions such as the time since desiccation or the past input of organic matter into the system may also influence CO₂ emissions^{38,39}"

13) Line 238: "small-scale" and insert comma after "patterns"

Done

14) Line 256: Insert comma after parenthesis

Done

15) Line 263: Replace "which" with "that"

Done

16) Line 269: Delete "for"

Done

17) Line 272: Insert comma after "matter"

Done

18) Line 278: Insert comma after "cycling"

Done

19) Line 310: "5 g" and delete "a"

Done

20) Line 320: Define "LOI"

We removed the abbreviation "LOI" and are using the term "organic matter" throughout the manuscript.

21) Supplementary Fig. 2: Typo in x-axis legend of bottom-left plot.

We removed this figure from the supplement due to the revised statistical approach.

Reviewer #3 (Remarks to the Author):

This manuscript reports a very nice datasets of 196 CO₂ surface-to-atmosphere fluxes measured from “dry” aquatic ecosystems – streams, reservoirs, lakes, or ponds that were either permanently or temporarily lacking surface water inundation. They find that CO₂ flux did not vary by climate zone but did vary by ecosystem type; CO₂ fluxes from dry streams and reservoirs were significantly lower than CO₂ fluxes from dry ponds. As we start to understand more about how aquatic ecosystems respond to drying under a changing climate this results is particularly important. They also compare these to measured adjacent hillslope fluxes; they do not present an analysis by region or ecosystem type, but find on average across the study that hillslope fluxes were higher than dry aquatic fluxes. Finally they use literature values to compare global average fluxes of aquatic ecosystems.

Some description and details are lacking that make evaluating the methods difficult. For example, How were field sites chosen?

All sites were chosen by the local teams who ensured that all sites were independent and not hydrologically connected in a direct upstream-downstream relationship. This information was added to the methods section (l. 313ff). The whole methods section has been revised to improve clarity.

Within field sites, how were replicate sampling locations chosen and how (was) that error propagated through the statistical analysis?

The three measurements per site were performed at slightly different spots and not at the very same spot three times. This means the chamber was inserted three times in different, nearby spots (typically few meters). This information was added to the methods section (l. 315-322). However, the three measurements per site are not replicates in our study. The different sites act as real replicates inside each ecosystem type (lake, reservoir, stream, pond). We designed the sampling to avoid pseudo-replication and have replicates that are truly independent. Therefore, we required the sites to be hydrologically independent and not connected to each other in a direct upstream-downstream connection.

How many sites were permanently dry versus temporarily, and any indication of how long they had been dry? Were these locations that typically dry seasonally or did they more recently dry?

The 196 sites cover a wide range of hydrological conditions but all of them have been temporarily dry. We did not collect the above-mentioned information systematically and, thus, also did not include it into the analysis. The lack of information (e.g. about the time of desiccation) is a trade-off of the multi-team approach because most sites are not monitored comprehensively enough to obtain this kind of information.

How were ecosystem types determined (ie a lake and pond are normally differentiated based on water depth, but for dry sites how was that distinction made)?

We extended the methods section to clarify the definition of the different ecosystem types. It now reads: “The sampling sites were classified into four inland water ecosystem types, based on the information provided by the local sampling teams. We defined “stream” as a natural watercourse that flows permanently or intermittently⁴⁸, “lake” as a naturally occurring low point in the landscape that contains standing water at least during certain periods⁴⁹, “reservoir” as a human made lake⁴⁹ and “pond” as a standing surface water body type that is considerably smaller than a lake or reservoir⁵⁰” (l. 321 - 326).

Was the “whole” lake, pond, reservoir dry or were measurements made on dry margins?

In cases where the whole system had dried up (e.g. small ponds, ephemeral streams), measurements were performed at representative parts of the bare sediment. In case of partial drying, measurements were performed at the emerged sediments at the shore. This information was added to the methods section (l. 318 - 321).

Which season were measurements taken and how was that incorporated into statistical analyses?

The date of the measurement was up to the local teams and all measurements were taken between 06.03.2016 and 11.10.2018. 38% of the measurements were taken in summer, 29% in spring, 22% in fall, and 11% in winter, respectively. However, all samplings were done during the dry season at sites with distinct dry and wet season. This may change depending on climate and hemisphere. Since we did not aim to disentangle seasonal patterns or characteristics we did not incorporate season into the statistical analysis.

The statistical analysis design was also difficult to follow – rather than using an approach that integrated the hierarchical nature of their predictor variables (proximal and distal) (perhaps something like structural equation modeling) they used three separate statistical models – LMM with proximal variables, followed by RF, followed by a second LMM, which seemed unnecessarily cumbersome.

We agree with the reviewer that the statistical analysis was hard to follow and also unnecessarily cumbersome. We therefore completely revised and simplified the statistical approach. As also explained in reply to Reviewer #1, we removed the iterative approach and base our findings solely on the linear mixed effects model (LMM). Initially, we chose the iterative 3-step approach to avoid model overfitting caused by the large number of predictors and their interactions in relation to our sampling size. By focusing on ecologically meaningful and relevant interactions *a-priori*, we have overcome this Problem from the first step. We therefore removed the variable selection by Random Forests and based the results on the Linear mixed-effects model. Consequently, we removed panel “b” from former Figure 4 (now Figure 3) showing details from the RandomForest analysis and updated the variables and estimates presented in former Figure 5 (now Figure 4) resulting from the LMM.

It is also unclear whether this approach was used for $n = 196$ dry aquatic ecosystem fluxes and also $n = 196$ adjacent uphill terrestrial fluxes – and if not, why?

We only analysed the dry aquatic ecosystems because the priority of the paper is to analyse the drivers of CO₂ emissions from dry inland waters and to evaluate the magnitude of these emissions in comparison to inundated water bodies and adjacent soils. An in-depth analysis of the environmental factors determining CO₂ emissions from soils was out of scope of this study.

Finally, the only reporting of model results is in two figures in the main text – the authors used AICc selection and averaging but there is no reporting of fit models in the main text or supplementary information.

We removed the AICc selection due to the revised statistical approach. We now show two figures (Figs. 3 and 4) with the results of the analyses of drivers using the LMM. In addition, we added the detailed results from the LMM to the Supplementary Table 2.

Finally, I found the argument about dry inland waters being included in global budgets difficult to follow and if I understand the authors correctly I disagree with it. The authors argue that current budgets likely underestimate CO₂ fluxes because they do not include dry inland waters – based on their data and how global budgets are produced it seems the opposite may be true. The authors are right – there is not a box for dry waters in global budgets and it is probably important to include one in global budgets especially with increased drying in the future in some regions; but at the scale and resolution of global budgets those dry water areas are included – either as terrestrial or aquatic ecosystems. Based on the comparison the authors present with uphill terrestrial fluxes (higher than dry water fluxes), global budgets that consider dry water land area as terrestrial are actually overestimating fluxes. Based on the comparison with inundated aquatic ecosystems (global averages), for streams again global budgets are overestimating fluxes. For lentic waters, global

averages were quite similar to measured dry fluxes (if you squint maybe you could argue for an underestimation of global fluxes for ponds). I understand the desire to place this dataset in a global context but I think there is important nuance that needs to be considered here, and would actually make the placement of the findings into a global context much richer.

We appreciate the reviewers' insightful comments of the difficulty to include CO₂ emissions from dry inland waters into global CO₂ budgets. We revised the discussion extensively to address the reviewers concerns (l. 262ff). We now focus our discussion on global inland water carbon budgets without considering terrestrial emissions. We discuss the complex relationships between flux magnitude and areal extent of dry inland waters in detail and address both, potential underestimation and potential overestimation of the impact of desiccated areas on global inland waters carbon inventory. The paragraph now reads:

“In any case, the net effect of including desiccated areas in current global inventories of carbon emissions from inland waters would depend on how desiccated areas have been considered in former studies, which is not always traceable. For instance, excluding CO₂ emissions from dry inland waters, as done in recent studies¹¹ would at first sight imply an underestimation of current inland waters CO₂ emissions to the atmosphere. However, the mistaken assignment of an intermittent stream as a permanent flow area may instead result in an overestimation of fluxes, as flowing waters appear to generally emit more CO₂ than the dry phases of intermittent rivers. On the contrary, dry areas of ponds, lakes, and reservoirs which global CO₂ flux assessments assigned wrongly as wetted areas would likely result in an underestimation of net fluxes. Recent global emission inventories have either disregarded desiccated areas^{11,42} (i.e., likely underestimating emissions) or incorporated intermittent streams using rough approaches, probably underestimating their area^{19,39} (i.e., likely overestimating emissions). Certainly, no current global estimate considers desiccated areas in ponds, lakes, and reservoirs, and thus these fluxes are likely to be underestimated. In sum, an assessment of the impact of desiccated areas on the global inland waters carbon inventory requires a much more accurate estimate of temporarily and permanently desiccated areas. Recent developments in remote sensing¹⁰ may help to incorporate desiccated areas from lakes, reservoirs, and large rivers, but an accurate estimate of intermittent stream and pond area is still a challenging endeavour considering most desiccated areas in vast regions of the world are obscured by cover (e.g. dense trees, clouds). This should be a research priority if CO₂ emissions from stream, rivers, and ponds are to be accurately incorporated into global inland water carbon flux estimates.”

Additional comments:

Ln 183 – 185 – Typo in this sentence, perhaps missing “or”.

Done

Ln 193 – Typo in this sentence. Delete “between”?

Done

Ln 109 – 111 – This is a prediction and not a hypothesis.

As already stated in reply to Reviewer #1, we agree that our hypotheses were not well defined in the original version of the manuscript. We restructured the introduction to be more precise. The essential section now reads (l. 112ff): “We hypothesize that CO₂ emissions from dry inland waters are above average compared to reported inland aquatic rates, thus making emissions from dry inland waters globally relevant. We further hypothesize that sediment-atmosphere emissions vary as a function of parameters controlling CO₂ production rates (such as organic matter supply, temperature, and moisture) and parameters controlling the transport of gas to the atmosphere (e.g., sediment texture) as well as geographical properties of the sampling locations which influence the biogeochemical conditions.”

Ln 107 – Permanently? At what point does a permanently dry area become terrestrial?

This is indeed a very interesting question. We are grateful for this impulse and think that this topic deserves its own studies. We rephrased the sentence to give less weight to this question here (l. 96-98): “In accordance with previous work¹², we define dry inland waters as the sections of lotic and lentic aquatic ecosystems on the Earth’s land masses in which surface water is absent, and sediments become exposed to the atmosphere.”

Figure 1 – This simple conceptual model is not that useful – it uses a lot of space to basically list 14 variables without much nuance about how those variables are related to one another. Also, colors are not readable by somebody that is colorblind.

We agree with the reviewer that Figure 1 is not particularly needed, especially, because of the revised statistical approach. We therefore removed it from the manuscript.

Ln 152 – The reporting of statistical results is confusing here – was the significance tested of the mean (lower) or variability (more variable) or both? I am also curious why the authors did not compare by ecosystem type or climate region. Also, this was a paired study – was that taken into account in statistical analysis?

We agree that this paragraph was somewhat confusing. We rephrased it to (l. 154-157): “CO₂ emissions from dry inland waters (mean = 186 mmol m⁻² day⁻¹) were in the same range, but significantly lower, than those from adjacent uphill soils which had not been previously inundated (mean ± SD = 222 ± 277 mmol m⁻² day⁻¹, median = 144, n = 196) (Wilcoxon signed rank test, P < 0.05)”. To take the paired characteristic into account, a Wilcoxon signed rank test was performed. The comparison of flux magnitude by ecosystem type and climate region is included in the following sections of the manuscript.

Ln 173 – 179 – If I am reading correctly there are no significant results in Figure 3a; yet the authors report pattern in that data in the following paragraph (polar regions lower, etc). This seems inappropriate.

We agree with the reviewer and changed the paragraph (l. 178ff) into “CO₂ emissions from polar (mean ± SD = 60 ± 58 mmol m⁻² day⁻¹, median = 36), continental (mean ± SD = 174 ± 140 mmol m⁻² day⁻¹, median = 125), temperate (mean ± SD = 178 ± 308 mmol m⁻² day⁻¹, median = 99), arid (mean ± SD = 233 ± 470 mmol m⁻² day⁻¹, median = 61) and tropical sites (mean ± SD = 236 ± 403 mmol m⁻² day⁻¹, median = 69) all fell within the same range (Fig. 3).”

Ln 179-181 – This is the only mention of the importance of season. How season was incorporated into experimental design is not included anywhere else in the manuscript. This is really important for gas fluxes.

We agree with the reviewer that seasonality is important for gas fluxes. However, it was not the aim of this study to reveal any information about seasonality of CO₂ emissions from dry sediments. We did not incorporate seasonality into the experimental design (we only measured once at every site during the dry season). The differentiation into sites having dry-summer, dry-winter or no dry season was done based on the Köppen-Geiger classification in order to provide a more in-depth analysis of the climate zone with the largest sampling size (temperate climate zone). We added this information to the methods section (l.352-354): “For an in-depth analysis of temperate sites, the 2nd order sub-groups dry-summer (Köppen-Geiger group Cs), dry-winter (Köppen-Geiger group Cw) and “without dry seasons” (Köppen-Geiger group Cf) were additionally distinguished.”

Ln 250 – 251 – That patterns by ecosystem type were consistent across climate zones is very interesting – those results were not presented in the text.

The sample size for the ecosystem types separated by climate zone is fairly low and so would be inappropriate to give this aspect too much weight (see figure below). We changed the sentence in

the text to be more precise (l. 248): “This pattern is consistent for most systems across all climate zones”

Ln 251 – The authors big take away in this paragraph is that dry aquatic fluxes are “a significant and globally prevalent source of CO₂ to the atmosphere” – although I find this paper quite interesting I’m not sure I agree. Line 255-256 the SD overlaps zero.

It is true that the SD overlaps zero but this does not mean that the difference from zero is, or is not, statistically significant. Our dataset is influenced by a huge variability caused by the large heterogeneity of the sampling sites which affects the conclusiveness of SD. Based on a Wilcoxon signed rank test, the CO₂ emission from dry inland waters is statistically different from zero, ($P < 2.2e-16$). For individual sites it was shown that dry inland waters are a significant source of CO₂ and we think that this is also relevant on a global scale¹.

Ln 335 – How was CO₂ flux log transformed if there were negative fluxes?

Values were log transformed by adding a constant to those variables showing negative values and/or values equal to zero. We rephrased in order to be more precise (l. 368): “We log-transformed CO₂ flux ($x + 28$), conductivity, organic matter content, moisture ($x + 0.1$) and elevation to meet the condition of normality and homogeneity of variance.”

In streams groundwater is an important source of CO₂ – in dry streambeds was groundwater discharge considered at all, or was in situ production considered dominant and why?

The reviewer is raising a good point here. Indeed, we think that groundwater may be an important source of CO₂ not only in streams but also in other systems. This is a complex process that is highly variable in time and space depending on the water level decline of the system. However, our study was not designed to capture impacts of groundwater discharge. To investigate the origin of CO₂ (*in situ* production vs. groundwater input) an intense study at a single site with high spatial and temporal resolution as well as more sophisticated methods like analysis of isotope signatures would be more appropriate. Nevertheless, CO₂ imported by groundwater is a potential source of uncertainty in our study. We therefore added this aspect into the discussion (l. 233).

REVIEWERS' COMMENTS:

Reviewer #1 (Remarks to the Author):

I greatly appreciate the efforts taken by the authors to address the concerns I had raised during the first review. I am satisfied that my concerns have now been addressed and am happy to see this manuscript published.

Reviewer #2 (Remarks to the Author):

This is a revised manuscript that presents the results of a large-scale, collaborative study designed to assess the relative importance of CO₂ emissions from inland waters during dry conditions. Net carbon dioxide emission rates from dry aquatic sediments and adjacent upland soils were measured using chambers and IRGAs at 196 sites in different ecosystem types and climatic zones. These data were complemented with measurements of proximal and distal correlates (i.e. potential drivers) of CO₂ emissions. I reviewed the original version and only have a few additional suggestions for further improvement of the manuscript.

- 1) Line 90: Two-thirds
- 2) Line 91: One-third
- 3) Lines 112 and 114: hypothesized
- 4) Line 112: "...above reported mean aquatic rates..."
- 5) Line 162: "...matter content, which would fuel CO₂ production, was greater..."
- 6) Line 186: Hot and dry? Not hot and wet?
- 7) Fig. 4 legend: The colons represent interactions, presumably?
- 8) Line 325: human-made
- 9) Line 361: 2nd-order

NCOMMS-19-6332178A – Reply Letter

REVIEWERS' COMMENTS:

Reviewer #1 (Remarks to the Author):

I greatly appreciate the efforts taken by the authors to address the concerns I had raised during the first review. I am satisfied that my concerns have now been addressed and am happy to see this manuscript published.

We are thankful for the reviewer's support and appreciate his/her efforts in reviewing our manuscript.

Reviewer #2 (Remarks to the Author):

This is a revised manuscript that presents the results of a large-scale, collaborative study designed to assess the relative importance of CO₂ emissions from inland waters during dry conditions. Net carbon dioxide emission rates from dry aquatic sediments and adjacent upland soils were measured using chambers and IRGAs at 196 sites in different ecosystem types and climatic zones. These data were complemented with measurements of proximal and distal correlates (i.e. potential drivers) of CO₂ emissions. I reviewed the original version and only have a few additional suggestions for further improvement of the manuscript.

We are thankful for the reviewer's support and appreciate his/her efforts in reviewing our manuscript.

1) Line 90: Two-thirds
done

2) Line 91: One-third
done

3) Lines 112 and 114: hypothesized
done

4) Line 112: "...above reported mean aquatic rates..."
done

5) Line 162: "...matter content, which would fuel CO₂ production, was greater..."
done

6) Line 186: Hot and dry? Not hot and wet?
Indeed, it's hot and wet conditions we are referring to. We changed the sentence accordingly.

7) Fig. 4 legend: The colons represent interactions, presumably?
The colons indeed represent interactions. We extended the figure legends for more clarity. It now reads: "Resulting coefficients from the linear mixed-effects model. Error bars indicate 95% confidence interval. Variables are shown in decreasing order of significance (analysis of variance, *** = $P < 0.001$, * = $P < 0.05$). Moisture, elevation and conductivity have been log₁₀-transformed

and all variables have been z-transformed prior to analysis. Colons indicate interaction between the respective variables.”

8) Line 325: human-made

done

9) Line 361: 2nd-order

done